# Automated Systems for Calculating Arteriovenous Ratio in Retinographies: A Scoping Review

**DOI:** 10.3390/diagnostics12112865

**Published:** 2022-11-18

**Authors:** Rosa García-Sierra, Victor M. López-Lifante, Erik Isusquiza Garcia, Antonio Heras, Idoia Besada, David Verde Lopez, Maria Teresa Alzamora, Rosa Forés, Pilar Montero-Alia, Jurgi Ugarte Anduaga, Pere Torán-Monserrat

**Affiliations:** 1Research Support Unit Metropolitana Nord, Primary Care Research Institut Jordi Gol (IDIAPJGol), 08303 Mataró, Spain; 2Multidisciplinary Research Group in Health and Society GREMSAS (2017 SGR 917), 08007 Barcelona, Spain; 3Nursing Department, Faculty of Medicine, Universitat Autònoma de Barcelona, Campus Bellaterra, 08193 Barcelona, Spain; 4Primary Care Group, Germans Trias i Pujol Research Institute (IGTP), 08916 Badalona, Spain; 5Palau-solità i Plegamans Primary Healthcare Centre, Palau-solità i Plegamans, Gerència d’Àmbit d’Atenció Primària Metropolitana Nord, Institut Català de la Salut, 08184 Barcelona, Spain; 6ULMA Medical Technologies, S. Coop, 20560 Onati, Spain; 7Primary Healthcare Centre Riu Nord-Riu Sud, Gerència d’Àmbit d’Atenció Primària Metropolitana Nord, Institut Català de la Salut, Santa Coloma de Gramenet, 08921 Barcelona, Spain; 8Institut Universitari d’Investigació en Atenció Primària Jordi Gol (IDIAP Jordi Gol), 08007 Barcelona, Spain; 9Department of Medicine, Faculty of Medicine, Universitat de Girona, 17004 Girona, Spain

**Keywords:** image processing, computer assisted, retinal vessels, retinal image analysis, diagnosis, hypertensive retinopathy, diagnostic screening programs

## Abstract

There is evidence of an association between hypertension and retinal arteriolar narrowing. Manual measurement of retinal vessels comes with additional variability, which can be eliminated using automated software. This scoping review aims to summarize research on automated retinal vessel analysis systems. Searches were performed on Medline, Scopus, and Cochrane to find studies examining automated systems for the diagnosis of retinal vascular alterations caused by hypertension using the following keywords: diagnosis; diagnostic screening programs; image processing, computer-assisted; artificial intelligence; electronic data processing; hypertensive retinopathy; hypertension; retinal vessels; arteriovenous ratio and retinal image analysis. The searches generated 433 articles. Of these, 25 articles published from 2010 to 2022 were included in the review. The retinographies analyzed were extracted from international databases and real scenarios. Automated systems to detect alterations in the retinal vasculature are being introduced into clinical practice for diagnosis in ophthalmology and other medical specialties due to the association of such changes with various diseases. These systems make the classification of hypertensive retinopathy and cardiovascular risk more reliable. They also make it possible for diagnosis to be performed in primary care, thus optimizing ophthalmological visits.

## 1. Introduction

Fundus photography, also known as retinography, is a popular imaging technique used to visualize changes in the retinal vessels through the pupil. It can capture changes in vascular caliber and the global geometric patterns of the retina [1]. It is also able to detect signs of retinopathy—such as microaneurysms, hemorrhages, cotton wool spots and hard exudates, and symptoms of the retinal arteriolar wall (e.g., generalized and focal arteriolar narrowing and arteriovenous nicking)—all of which are often observed in patients with systemic diseases, such as diabetes and hypertension.

Regarding diabetes, the 1989 Saint Vincent Declaration set the goal of reducing diabetes-related blindness by one-third over the next five years. This was restated in the 2005 Liverpool Declaration’s objective of establishing systematic screening programs to reach at least 80% of the population with diabetes by 2021 [2]. The increased demand for diabetic retinopathy (DR) screening resulting from systematic programs could be met using automated retinal image analysis systems. Such systems can be used in different DR screening scenarios and offer relatively high sensitivity and a substantial reduction in the workload of the health system. Moreover, they are now mature enough to be safely used in DR screening [3,4]. Automated tools improve the quality of DR screening and accessibility to medical care while reducing the cost of the disease by promoting early detection and treatment, which is essential to stop progression [5].

Regarding hypertension, there is evidence that it is associated with retinal arteriolar narrowing. Retinal vessel diameter is expressed as arteriovenous ratio (AVR). According to the Keith–Wagener–Barker classification, AVR values of less than 0.66 reflect hypertensive retinopathy [6].

Arteriolar narrowing is associated with more severe coronary heart disease, stroke, and mortality [6,7,8,9]. There has also been a recent increase in evidence showing that retinal arteriolar narrowing, retinal venular widening, and a suboptimal retinal vascular network are associated with poorer cognitive performance [10,11,12]. Retinal imaging techniques provide unique information about the state of the microvasculature and neuronal structure, different from current neuroimaging markers, such as brain magnetic resonance imaging (MRI), and systemic markers, such as blood pressure. While retinal imaging cannot fully replace PET scans or MRIs in the diagnosis of disease, it does offer a complementary approach to these brain imaging techniques and has considerable potential in clinical and research settings [13].

For all these reasons, retinal imaging can be used as a risk stratification tool because studies suggest that the addition of retinal measures improves the prediction of stroke (an improvement on established risk factors of approximately 10%) [14,15]. Although just a modest improvement in prediction, these findings suggest that adding a combination of various retinal features and/or retinal functional parameters (i.e., a “multimarker approach”) may further improve the prediction of dementia and stroke. It might also enable the identification of a more specific subgroup of patients who could benefit from more intensive and expensive examinations, such as brain MRI.

Retinal vascular imaging has also been used to examine the effects of antihypertensive therapy, showing that lowering blood pressure leads to the regression of retinal vascular signs [16,17]. While there have been no significant intervention studies using changes in retinal images as alternative outcome measurements in dementia and stroke, this approach has substantial potential. In addition to its clinical value, retinal imaging may also be a worthwhile research tool in major brain and neurological diseases, such as multiple sclerosis [18,19,20], depression [21,22], and schizophrenia [23,24].

Evaluating retinographies manually implies additional variability in retinal vessel measurements, even when following a standardized protocol. This variability is eliminated if fully automated software is used to measure retinal vascular caliber and other anomalies, although there may be other additional sources of variation in the measurements, such as retinal pigmentation, pupil dilation, the presence of cataracts and other media opacities, photographic technique, type of camera (mydriatic/non-mydriatic or desktop/portable), and image quality (brightness, focus, and contrast) [25]. Manually segmenting vessels, labeling arteries and veins, and localizing the optic disc is a time-consuming task that decreases process efficiency. However, over the past two decades, multiple software systems have been developed to measure and semi-automatically assess the retinal vessel caliber from fundus photographs using artificial intelligence (AI) algorithms [26].

Research question: What is the current stage of implementation of automated retinal vessel analysis systems retinographies?

Aim: This scoping review aims to summarize the research available on automated retinal vessel measurement systems so they may be considered in future research and introduced into clinical practice.

## 2. Materials and Methods

This review followed the PRISMA extension checklist for scoping reviews [27].

### 2.1. Search Strategy, Data Sources, and Selection

Searches were performed in the Medline, Scopus, and Cochrane electronic databases to locate studies published between 1 january 2004 and 1 september 2022 examining automated systems for the diagnosis of retinal vessel alterations caused by hypertension. The following keywords were used: diagnosis; diagnostic screening programs; image processing, computer-assisted; artificial intelligence; electronic data processing; hypertensive retinopathy; hypertension; retinal vessels; arteriovenous ratio (no MeSH); retinal image analysis (no MeSH).

### 2.2. Selection Criteria

Articles were included in the review if they met the following inclusion criteria:(1)Automated systems were used to partially or totally analyze photographic images of the retina.(2)Changes in the retinal vascular network and/or retinal vascular measurements were analyzed.(3)The publication was peer-reviewed.(4)The study was observational, descriptive (population, cross-sectional), analytical (case studies and controls, cohorts), experimental (clinical trials), or a validation of experiments/new image analysis methods.

Only papers written in English were selected. Studies using automated systems for diabetes screening were excluded. Qualitative studies and gray literature were excluded.

### 2.3. Selection of Studies

Abstracts and articles were independently reviewed by two members of the research team based on predetermined inclusion and exclusion criteria. When it was unclear whether an article should be included or some discrepancy appeared, the coordinating researcher of the study also reviewed it.

### 2.4. Data Extraction

Initial data extraction elements included: author, country, research aim, study design, study setting, interpretation system, degree of software automation (semi-automatic, automatic), lesions that the system was able to detect, focus of the photograph and area analyzed, sensitivity, specificity, diagnostic precision, economic evaluation, time savings, local management or the possibility of electronically sending the image to a repository, possibility of comparison for patient follow-up.

Data from each article were independently extracted by two of the authors and then verified by two others.

## 3. Results

### 3.1. Search Process

The PRISMA flow diagram (Figure 1) describes the steps taken to select the articles [28]. The search strategies generated 433 articles, of which 58 full-length articles were evaluated for eligibility. Of these, 25 articles were included in the scoping review.

### 3.2. Characteristics of the Articles

The articles included in the review were published between 2010 and 2022, and the number of retinographies analyzed ranged from 20 to 95,716. These retinographies were provided by international databases, as well as real scenarios. Table 1 presents a summary of the characteristics of the studies, including first author, year of publication, the country where the study was conducted, study aim, sample, number of retinographies, name of the software used, degree of automation, and, lastly, the scenario in which it was tested.

The median number of retinographies analyzed in the included studies was 180, with a maximum of 54,714 and a minimum of 40. In relation to the degree of automation, in 10 articles they used automatic software and in 15 semi-automated systems.

### 3.3. Interpretation Procedures

The analysis systems included in these articles detect various alterations of the vessels, including tortuosity, arteriolar and venular caliber, and even AVR calculation. Table 2 shows a description of the retinal lesions detected in each study, the focus of the photograph, and the area of the retina that was analyzed. The focus of the image was mainly on the macula and optic disc, and the analyzed area ranged between 2 to 3 radii from the optic disc.

Lastly, the sensitivity, specificity, and diagnostic accuracy were recorded, although this information was only reported in seven articles.

The researchers were initially interested in studying additional data, such as economic evaluation, time savings, local management or the possibility of electronically sending the image to a repository, and the possibility of comparison for patient follow-up. However, the articles selected did not provide this information; therefore, such data were not collected.

### 3.4. Summary of the Results

Most of the articles reviewed fall into two main categories. The first includes publications dealing with automated or semi-automated systems that measure retinal vessels as a diagnostic method for other pathological processes. Retinal vessel measurement can be useful for diagnosing pathologies related to cardiovascular risk [26,34,41] and hypertension [26,33,34,36], dementia and stroke [35], glaucoma [37], chronic kidney disease [11], glycemic control in children [40], myopia [39], and severity of diabetic macular edema [45].

The second category comprises articles aimed at developing automated retinal vessel measurement systems. These articles cover several levels of development: those that use systems limited to vessel segmentation [46,49,51]; those that include vessel labeling and creation of the vascular tree [31,42,45,47,52]; and those that calculate AVR in order to grade hypertensive retinopathy [43,44,49,53].

## 4. Discussion

This scoping review aimed to summarize the research available on automated retinal vessel analysis systems in order to determine where automated AVR calculation systems are currently at in terms of implementation. The results indicate that interest in developing technology that facilitates the analysis of the retinal microvascular network has increased over the past eight years. Publications from 2014 and earlier refer to experiments to test the algorithms developed. As of 2015, the software developed from these algorithms has been introduced into clinical practice, demonstrating advantages in real-life scenarios, even though it is not yet widely nor systematically employed. The results of this review confirm that automated AVR calculation systems have not just been introduced as a diagnostic tool for retinal vascular disorders in the field of ophthalmology, but they have been extensively applied in other medical specialties as an accessible and efficient diagnostic tool for other pathologies. Numerous findings indicate that retinal vascular caliber is associated with various systemic diseases, such as hypertension, obesity, diabetes, chronic kidney disease, and stroke. Moreover, since AVR is associated with the development of cardiovascular disease, examining it in this way offers a non-invasive view into the systemic microvasculature.

Hypertensive retinopathy is an indicator of damage to other target organs. However, it is difficult for ophthalmologists to study hypertensive retinopathy in the early stages. Therefore, further research should be conducted on computer-assisted diagnoses that use AVR calculation to automatically detect hypertensive retinopathy and grade it in its early stages [54].

The automation of processes is a prerequisite to improving the affordability, efficiency, and accessibility of these procedures [55] and reducing the high subjectivity of manually assessing AVR [56]. Deep learning methods arise to compute AVR. Convolutional Neural Networks (CNN, Atlanta, GA, USA) obtain a good approximation of AVR value by applying a sequence of spatial filters, subsampling, and non-linear operations.

The clinical applications of artificial intelligence in automated AVR reading could cover a wide range of tasks, including automating hypertensive retinopathy screening, supporting treatment decision-making, assessing systemic vascular status and cardiovascular mortality [57], prescribing medications and diagnostic tests, and creating prognostic models of different diseases [58] to provide more efficient, precise, and sensitive methods in the interpretation of clinical data.

Nonetheless, using artificial intelligence to analyze retinal microvasculature does present some limitations. Firstly, the images used to validate and train the AI may not have enough ethnic variability to provide high external validity because there must be enough, but not too many, images for the processes to be efficient. Secondly, the data should be restricted to those criteria of greatest prognostic relevance, maintaining maximum diagnostic accuracy and minimum algorithm complexity. Consequently, if these processes were automated, they could be implemented in primary care for use by trained health professionals and in rural settings, thus facilitating the classification of cardiovascular risk and reducing the need to refer cases for evaluation by an ophthalmologist. Overall, this would result in the optimization of available health resources.

## 5. Conclusions

In recent years, there has been increased interest in developing technology that facilitates the analysis of the retinal microvascular network. Software has been developed and is being introduced into clinical practice not just as a diagnostic tool in the field of ophthalmology, but also in other medical specialties, because there is an established association between various diseases and retinal vessel alterations.

If automated processes for retinal vessel measurement were implemented in primary care for use by trained health professionals, fewer cases would need evaluation by an ophthalmologist, thus optimizing the available health resources.

Moreover, such processes improve the reliability of vasculature measurements, which, in turn, leads to better classification of hypertensive retinopathy by eliminating observer subjectivity and taking cardiovascular risk into account. Also, the more reliable the measurements, the better the early diagnosis of other pathologies, such as dementia and stroke. Further research on the evaluation and implementation of these technologies is needed to recommend their use.

## Figures and Tables

**Figure 1 diagnostics-12-02865-f001:**
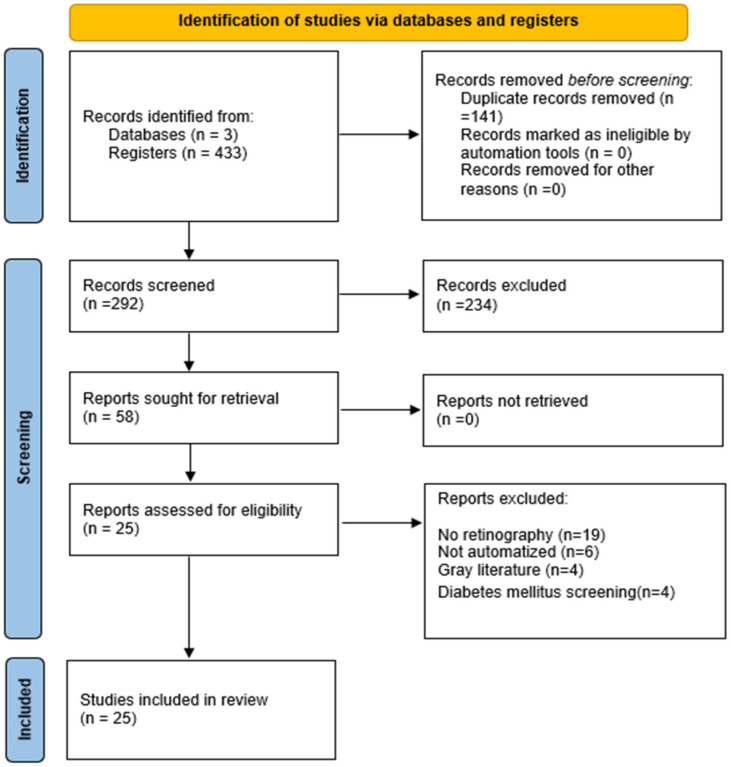
PRISMA 2020 flow diagram.

**Table 1 diagnostics-12-02865-t001:** Characteristics of the studies.

Author, Year	Country	Aim	Sample	Photographs	Software	Automation Type	Scenario
Badawi, 2022 [29]	Pakistan and United Arab Emirates	Measure arteriovenous relationship and degree of retinal vessel tortuosity to detect and classify hypertensive retinopathy.	504	504	VAMPIRE ^a^	Automated	RVM dataset ^b^
Huang, 2022 [30]	China	Assess the association between cumulative blood pressure, averaged over 25 years, and retinal vessel calibers.	1818			Semi-automated	ARIC ^c^
Irshad, 2021 [31]	Australia	Method for the differentiation and classification of retinal vessels by Binary Particle Swarm Optimization (BPSO).		142	BPSO	Automated	Real scenario INSPIRE-AVR VICAVR
Dai, 2020 [32]	China	Report on the construction of a model to further explore the pathophysiological changes of the retinal microvasculature.	1419	2012	CNN ^8^ architecture	Automated	DRIVE ^1^ STARE ^2^
Maderuelo-Fernandez, 2020 [26]	Spain	Assess the relevance of a retinal vessel analysis system in target organ damage and vascular risk.	250	495	ALTAIR ^d^	Semi-automated	Real scenario
Robertson, 2020 [33]	UK	Evaluate whether retinal vessel measurements were associated with hypertension.	440	880	VAMPIRE ^a^	Semi-automated	NICOLA ^3^
Tapp, 2019 [34]	UK	Examine the association between retinal vessel morphometry and blood pressure and arterial stiffness.	54,714	95,716	QUARTZ ^e^	Automated	United Kingdom Biobank
Lau, 2019 [35]	Hong Kong	Determine whether a high burden of white matter hyperintensities can be detected through images of the retina.	180	180	ARIA ^f^	Automated	CU-RISK COHORT ^4^
He, 2018 [36]	China	Examine the association between blood pressure measurements and changes in the retinal microvasculature.	1501	1501	IVAN ^g^	Semi-automated	Real scenario Pediatrics
Adiarti, 2018 [37]	Indonesia	Assess retinal vessel diameter as a marker of glaucoma.	54	108	SIVA ^h^	Semi-automated	Real scenario
Akbar, 2018 [38]	Pakistan	Assess AVR calculation as a classifier of the degree of arterial hypertension.	100	198	-	Automated	INSPIRE-AVR ^5^ VICAVRAVRDB ^6^
Iwase, 2017 [39]	Japan	Compare a new method of retinal vessel measurement with the IVAN system.	99	180	-	Semi-automated	Real scenario
Yip, 2017 [11]	Singapore	Describe the associations between retinal vascular parameters and chronic kidney disease.	1256	2512	SIVA ^h^	Semi-automated	Real scenario
Li, 2017 [40]	Singapore	Investigate the association between poor glycemic control and subsequent changes in the retinal microvasculature.	55	110	SIVA ^jh^	Semi-automated	Real scenario Pediatrics
Vázquez Dorrego, 2016 [41]	Spain	Evaluate the usefulness of measuring arteriovenous ratio to detect silent brain ischemia.	768	2262	VesselMap2	Semi-automated	Real scenario
Cavallari, 2015 [42]	USA, Italy	Develop a semi-automated method to assess retinal vessel morphology.	54	108	Cioran and BRetina plugins	Semi-automated	Real scenario
Fraz, 2015 [43]	Pakistan and UK	Present a fully automated software to analyze the retinal vasculature.	-	16,000	QUARTZ ^e^	Automated	DRIVE ^1^ STARE ^2^ INSPIRE ^5^ CHASE-DB1 DIARETDB1 ^7^
Estrada, 2015 [44]	USA	Develop a semi-automated method to distinguish arteries from veins in fundus images.	110	130	-	Semi-automated	DRIVE ^1^ INSPIRE ^5^ WIDE
Moradi, 2014 [45]	USA	Identify the association between baseline retinal vascular caliber and visual outcome of patients with diabetic macular edema.	84	25	IVAN ^g^	Semi-automated	Real scenario
Franklin, 2014 [46]	India	Present an automated retinal vessel segmentation technique.	40	40	ANN ^9^	Automated	DRIVE ^1^
Dashtbozorg, 2014 [47]	Portugal	Develop an automatic approach for the classification of arteries and veins of the retinal vasculature.	-	130	-	Semi-automated	DRIVE^1^ INSPIRE-AVR ^5^ VICAVR
Vázquez, 2013 [48]	Spain	Propose a methodology for classifying arteries and veins in the fundus vasculature.	-	100	-	Semi-automated	VICAVR-2
Huang, 2012 [49]	China	Propose an automated computational framework for retinal vascular network labeling and branch order analysis.	-	40	-	Automated	DRIVE ^1^
Ortega, 2010 [50]	Spain	Develop a generic framework for processing retinal images.	96	173	SIRIUS ^i^	Semi-automated	Real scenario
Villalobos-Castaldi, 2010 [51]	Mexico	Present a fast, efficient, and automatic algorithm to extract vessels from retinal images.	-	20	-	Automated	DRIVE ^1^

VAMPIRE ^a^: Vessel Assessment and Measurement Platform for Images of the REtina. RVM b: Retinal Vessel Morphometry. ARIC c: Atherosclerosis Risk in Communities. ALTAIR d: Automatic image anaLyzer to assess reTinAl vessel calIber. QUARTZ e: Quantitative Analysis of Retinal Vessel Topology and size. ARIA f: Automated Retinal Image Analyzer (an open-source software designed for automatic recognition and computation of retinal factors and parameters). IVAN g: Interactive Vessel Analysis, University of Wisconsin–Madison. SIVA h: Singapore “I” Vessel Assessment. SIRIUS i: System for the Integration of Retinal Images Understanding Services. DRIVE 1: Digital Retinal Images for Vessel Extraction. STARE 2: STructured Analysis of the Retina. NICOLA 3: Northern Ireland COhort for the Longitudinal study of Ageing. CU-RISK COHORT 4: The Chinese University of Hong Kong—Risk index for Subclinical Brain Lesions in Hong Kong. INSPIRE-AVR 5: Iowa Normative Set for Processing Images of the REtina-ArterioVenous Ratio. AVRDB 6: local dataset Annotated dataset for Vessel Segmentation and Calculation of Arteriovenous Ratio. DIARETDB1 7: Standard DIAbetic RETinopathy Database caliBration level 1. CNN 8: Convolutional Neural Networks. ANN 9: Artificial Neural Networks. BPSO: Binary Particle Swarm Optimization.

**Table 2 diagnostics-12-02865-t002:** Lesions detected.

Author, Year	Measurements	Focus of the Image	Area Analyzed	S *	SP **	DP ***	Conclusions
Badawi, 2022 [29]	AVR Tortuosity	Optic disc	2 to 3 radii from the optic disc The entire retina	95.5%	-	96.8%	Hybrid tool that combines AVR and tortuosity to detect and grade the severity of hypertensive retinopathy.
Huang, 2022 [30]	CRAE ^a^ CRVE ^b^ AVR	Optic disc	2 to 3 radii from the optic disc	-	-	-	High blood pressure, averaged over 25 years, and specifically DBP, was associated with narrower retinal vessel diameter.
Irshad, 2021 [31]	Classification of arteries and veins	Optic disc	2 to 3 radii from the optic disc	-	-	92.7% 94.6% 91.9%	Proposal of a method that offers improved retinal vessel classification and is robust in three different databases.
Dai, 2020 [32]	Subclinical morphological features	Macula	The entire retinal vasculature	59.3%	63.8%	60.9% 70.5% AUC: 65.1%	Changes in retinal vessel branching pattern were the most significant response to high blood pressure compared to other retinal microvascular biomarkers such as caliber, tortuosity, fractal dimension, and branching angle.
Maderuelo-Fernandez, 2020 [26]	CRAE ^a^ CRV ^b^ AVR	Optic disc	Three concentric circles around the optic disc	-	-	-	A concomitant association of retinal vessel measurements with other cardiovascular parameters and cardiovascular risk is shown.
Robertson, 2020 [33]	Nasal-annular AVR	Annular segment that subtends 180° nasally to the optic disc	6.5 to 8.5 radii from the optic disc	-	-	-	Semi-automated AVR measurements on ultra-widefield fundus images were associated with hypertension.
Tapp, 2019 [34]	Arteriolar and venular diameter and Tortuosity	Optic disc and macula	The entire retinal vasculature	-	-	-	Associations between retinal vessel morphometry, blood pressure, and arterial stiffness index.
Lau, 2019 [35]	CRAE ^a^ CRVE ^b^ Arteriole occlusion Hemorrhages Tortuosity	Macula	2 to 3 radii from the optic disc	93%	98%	-	Automatic retinal image analysis can detect community-dwelling subjects who do not have dementia and who have a significant burden of white matter hyperintensities in their brains.
He, 2018 [36]	CRAE ^a^ CRVE ^b^ AVR	Optic disc and macula		-	-	-	Higher blood pressure was significantly associated with narrower retinal arterioles in a population of 12-year-olds.
Adiarti, 2018 [37]	CRAE ^a^ CRVE ^b^ AVR	Optic disc and macula	1 to 4 radii from the optic disc	-	-	-	Retinal arteriolar narrowing may represent subclinical microcirculatory changes associated with the presence of a glaucomatous optic disc even in the absence of increased intraocular pressure.
Akbar, 2018 [38]	CRAE ^a^ CRVE ^b^ AVR	Optic disc	The entire retinal vasculature	98.9%	98.6%	98.8%	The system is reliable for clinical use in the detection and grading of hypertensive retinopathy.
Iwase, 2017 [39]	CRAE ^a^ CRVE ^b^ AVR	Optic disc	2 to 3 radii from the optic disc	-	-	-	The method would be especially useful to accurately measure retinal vessel caliber in a myopic population.
Yip, 2017 [11]	CRAE ^a^ CRVE ^b^ Tortuosity	Optic disc	1 to 4 radii from the optic disc	-	-	-	Retinal microvascular abnormalities may reflect early subclinical damage to the renal microvasculature that is later associated with the development of chronic kidney disease.
Li, 2017 [40]	CRAE ^a^ CRVE ^b^ Tortuosity	Optic disc and macula	1 to 4 radii from the optic disc	-	-	-	Pediatric patients with Type 1 diabetes and poor glycemic control showed abnormal retinal morphology in the short term.
Vázquez Dorrego, 2016 [41]	AVR	Optic disc and macula	2 and 3 radii from the optic disc	-	-	-	Alteration of the retinal vasculature is associated with an increased risk of silent brain ischemia in hypertensive patients.
Cavallari, 2015 [42]	AVR Tortuosity Mean Fractal Dimension	Optic disc	3.5 radii from the optic disc	68.8% (HR) 54.5% (CADASIL)	87.5% (HR) 90.9% (CADASIL)	-	AVR, tortuosity index, and mean fractal dimension were altered in HR and CADASIL subjects compared to age- and sex-matched control subjects.
Fraz, 2015 [43]	AVR Tortuosity	Optic disc	The entire retinal vasculature	75.5%	98.0%	95.3%	Provides quantifiable measurements of retinal vessel morphology.
Estrada, 2015 [44]	Classification of arteries and veins	Optic disc and macula	The entire retinal vasculature	91.0% 93.0% 91.7% 91.5%	91.0% 94.1% 91.7% 90.2%	90.9% 93.5% 91.7% 90.9%	The software outputs a graph representing the retinal vasculature.
Moradi, 2014 [45]	CRAE ^a^ CRVE ^b^	Optic disc	2 and 3 radii from the optic disc	-	-	-	Correlation between retinal venular caliber and visual outcome in patients with diabetic macular edema treated with ranibizumab. A higher CRVE, but not CRAE, was correlated with an improvement in vision.
Franklin, 2014 [46]	Vessel segmentation	Macula	The entire retinal vasculature	-	-	-	This technique has proven to be an effective tool for blood vessel segmentation in retinal images.
Dashtbozorg, 2014 [47]	Classification of arteries and veins	Optic disc	The entire retinal vasculature	91% 90%	86% 84%	-	The software outputs a graph representing the retinal vasculature. Each segment of the retina is then classified as an artery or vein.
Vázquez, 2013 [48]	Classification of arteries and veins	Optic disc	Various circumferences around the optic disc	-	-	87.7%	The best results were achieved with four separate circumferences with a value of 0.5 radii.
Huang, 2012 [49]	Skeleton of the retinal vascular tree	Optic disc	The entire retinal vasculature	-	-	-	A useful tool to extract morphological characteristics in pathological studies related to the retina.
Ortega, 2010 [50]	AVR	Optic disc	Various circumferences around the optic disc	-	-	99.2%	Sirius implements a web-based solution to analyze, manage, and understand retinal images.
Villalobos-Castaldi, 2010 [51]	Vessel segmentation	Optic disc	The entire retinal vasculature	96.5%	94.8%	97.6%	Tool to obtain an automatic threshold value to segment vessels.

S * Maximum sensitivity. SP ** Maximum specificity. DP *** Diagnostic precision. CRAE ^a^: Central Retinal Arteriolar Equivalent. CRVE ^b^: Central Retinal Venular Equivalent. AVR: Arteriole–to–Venule Ratio. CADASIL: Cerebral Autosomal Dominant Arteriopathy with Subcortical Infarcts and Leukoencephalopathy. HR: Hypertensive Retinopathy.

## Data Availability

Not applicable.

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
