# Peer review of "Automated Systems for Calculating Arteriovenous Ratio in Retinographies: A Scoping Review"

_diagnostics, 2022, doi:10.3390/diagnostics12112865_

Round 1
Reviewer 1 Report
This interesting systematic review covers the interesting topic of automated systems used in retinal vascular diseases.
1) Authors need to expand the results section. Data are presented mainly in tables (which is acceptable but not sufficient).
2) Please check for typos i.e. QUART is written as QUARTZ.
3) Could you please check whether any studies on AngioTool could be included?
In conclusion, it's a nice article but some minor revisions would further improve the manuscript before final acceptance.
Author Response
Thank you for your suggestions.
Point 1. Authors need to expand the results section. Data are presented mainly in tables (which is acceptable but not sufficient).
Response 1: Some information have been added at 3.2 and 3.3 sections.
Point 2. Please check for typos i.e. QUART is written as QUARTZ
Response 2:The typos have been adressed
Point 3. Could you please check whether any studies on AngioTool could be included?
Response 3: There are studies conducted on AngioTool based on OCT images. Nevertheless, the present study sets retinal images as inclusion criteria, which means the exclusion of OCT images.
Reviewer 2 Report
Rosa Garcia-Sierra et al. presented their scoping review regarding calculating arteriovenous ratio in fundus.
major flaws need to be addressed before further consideration
- did you perfom a declaration to the prospero website ?
- Introduction: authors should present qualitative (nicking, exudate...) and quantitative retinal vascular biomarkers (tortuosity, caliber, fractal dimension...)
- Introduction: authors should present semi automated software ==> SIVA IVAN VAMPIRE...and the proccess for migration to automated algorithms
- Introduction: in the title you restricted your review to "calculating AVR" in the aim : "summarize available on automated retinal vessel analysis" the topic is not the same please be consistent
- Introduction: researchs on depression and schizophrenia are very controversial you should not cite them
- Introduction: https://doi.org/10.1371/journal.pone.0194694 this paper could be discussed
- Selection criteria: language ?
- Authors should be more clear about their strategy to include or exclude articles (how many authors were involved ? in case of discrepancy ?)
- Results: semi automated and automated strategy are both included this is not the same...it has to be modified
- very surprised a lot of paper are missing ==> carol y cheung SIVA-DL 2020, automorph tvst 2022...the research algorithm is not reliable
- Main results shoul be presented
- discussion is too vague and not scientifically impactfull
perspective should be discussed / issued with automatised algorithm should be discussed
Author Response
Thank you for your suggestions, all have been very useful for the improvement of the manuscript.
Point 1. did you perfom a declaration to the prospero website ?
Response 1: No, I didn’t do it, because PROSPERO does not accept scoping reviews
Point 2. Introduction: authors should present qualitative (nicking, exudate...) and quantitative retinal vascular biomarkers (tortuosity, caliber, fractal dimension...)
Response 2: The first paragraph (line 42 to 49) of the introduction presents the changes that can be captured by a retinography explained as changes in calibre and geometric patterns, microaneurysms, haemorrhages, cotton wool spots and hard exudates, signs of the retinal arteriolar wall (generalized and focal arteriolar narrowing and arteriovenous nicking). Our opinion is that a more comprehensive explanation could difficult the reading of the manuscript.
Point 3. Introduction: authors should present semi automated software ==> SIVA IVAN VAMPIRE...and the proccess for migration to automated algorithms
Response 3: We have introduced a paragraph with this information between lines 98 and 100.
Point 4. Introduction: in the title you restricted your review to "calculating AVR" in the aim : "summarize available on automated retinal vessel analysis" the topic is not the same please be consistent
Response 4: At the aim, we have introduced the word measurement, in order to be more consistent with the title.
Point 5. Introduction: researchs on depression and schizophrenia are very controversial you should not cite them
Response 5: The research cited in the manuscript is published in high impact journal, some of them, the related with stroke, has been performed by our group. We consider that the topic of this review is interesting for this field of study. Accordingly, we prefer to maintain this cites.
Point 6. Introduction: https://doi.org/10.1371/journal.pone.0194694 this paper could be discussed
Response 6: Due to the relevant conclusions of the paper this paper, we have added in the discussion section.
Point 7. Selection criteria: language ?
Response 7: Only papers written in English were selected. Has been added at the selection criteria
Point 8 Authors should be more clear about their strategy to include or exclude articles (how many authors were involved ? in case of discrepancy ?)
Response 8: at 2.3 section (Selection of studies) is explained the number of authors involved in the selection. We have added a clarification about unclear articles.
Point 9. Results: semi automated and automated strategy are both included this is not the same...it has to be modified
Response 9: The degree of automation is shown in table 1
Point 10. very surprised a lot of paper are missing ==> carol y cheung SIVA-DL 2020, automorph tvst 2022...the research algorithm is not reliable
Response 10: the paper ‘Comparison of Common Retinal Vessel Caliber Measurement Software and a Conversion Algorithm’ was excluded at the title and abstract review. There was agreement that it did not meet the inclusion criteria.
Point 11. - Main results shoul be presented
Response 11: the results written apart from the tables have been expanded
Point 12. discussion is too vague and not scientifically impactfull
Response 12: Discussion has been expanded
- Point 13. perspective should be discussed / issued with automatised algorithm should be discussed
Response 13: Sorry, but we don’t understand this comment
Reviewer 3 Report
I thank the editors for the opportunity to review this article.
The article as a scoping review, examined the current literature that has been published in the field of AI for calculating arterial venous ratios. This is a predominantly narrative review.
I would have liked the authors to have discussed limitations of the various systems and whether these serve as alternatives to achieving a diagnosis or as a clinical decision support tool. For instance, the authors describe an arterial venous ratio of less than 0.2 that is suggestive of malignant retinopathy. However, there are other signs and symptoms that come into the forefront for consideration when making this diagnosis.
I do agree that the use of computer systems allow allows for removal or subjectivity and inter examiner variability. I would have liked the authors to discuss some of the machine learning techniques that are used to train the automated systems as these may differ in the ability to continuously adapt using large data sets of information and also learning through ongoing analyses.
The authors conclude that such processes improves reliability of vascular measurements and specifically mentioned classification of hypertensive retinopathy. However, within the paper, the author's discuss a range of studies in a variety of diseases and the work that has been done to seek to understand the correlation between retinal vasculature retinal architecture and other diseases such as dementia, Alzheimer's disease amongst others.
I would have liked for the authors to keep the conclusion broad to maintain congruence with the main text of the article.
Author Response
Thank you for your suggestions, all three have been very useful for the improvement of the manuscript.
Point 1. I would have liked the authors to have discussed limitations of the various systems and whether these serve as alternatives to achieving a diagnosis or as a clinical decision support tool. For instance, the authors describe an arterial venous ratio of less than 0.2 that is suggestive of malignant retinopathy. However, there are other signs and symptoms that come into the forefront for consideration when making this diagnosis.
Response 1: By agreement of the authors, we have eliminated the phrase that referred to the AVR value of 0.2, since this value has little significance among the general population.
Point 2. I do agree that the use of computer systems allow allows for removal or subjectivity and inter examiner variability. I would have liked the authors to discuss some of the machine learning techniques that are used to train the automated systems as these may differ in the ability to continuously adapt using large data sets of information and also learning through ongoing analyses.
Response 2: A little explanation of the different machine learning techniques has been added in the discussion section
Point 3. The authors conclude that such processes improves reliability of vascular measurements and specifically mentioned classification of hypertensive retinopathy. However, within the paper, the author's discuss a range of studies in a variety of diseases and the work that has been done to seek to understand the correlation between retinal vasculature retinal architecture and other diseases such as dementia, Alzheimer's disease amongst others.
I would have liked for the authors to keep the conclusion broad to maintain congruence with the main text of the article.
Response 3: A sentence to maintain the congruence with the main text has been added in the conclusions.